# Homophily and social mixing in a small community: Implications for infectious disease transmission

**Dana K. Pasquale**[1,2,3]*, **Whitney Welsh**[4], **Keisha L. Bentley-Edwards**[5], **Andrew Olson**[6], **Madelynn C. Wellons**[2,3], **James Moody**[2,3]

**1** Department of Population Health Sciences, Duke University School of Medicine, Durham, North Carolina, United States of America, **2** Department of Sociology, Duke University, Durham, North Carolina, United States of America, **3** Duke Network Analysis Center, Social Science Research Institute, Duke University, Durham, North Carolina, United States of America, **4** Social Science Research Institute, Duke University, Durham, North Carolina, United States of America, **5** Samuel DuBois Cook Center on Social Equity, Duke University, Durham, North Carolina, United States of America, **6** Duke AI Health, Duke University School of Medicine, Durham, North Carolina, United States of America

\* dana.pasquale@duke.edu

## Abstract

Community mixing patterns by sociodemographic traits can inform the risk of epidemic spread among groups, and the balance of in- and out-group mixing affects epidemic potential. Understanding mixing patterns can provide insight about potential transmission pathways throughout a community. We used a snowball sampling design to enroll people recently diagnosed with SARS-CoV-2 in an ethnically and racially diverse county and asked them to describe their close contacts and recruit some contacts to enroll in the study. We constructed egocentric networks of the participants and their contacts and assessed age-mixing, ethnic/racial homophily, and gender homophily. The total size of the egocentric networks was 2,544 people (n = 384 index cases + n = 2,160 recruited peers or other contacts). We observed high rates of in-group mixing among ethnic/racial groups compared to the ethnic/racial proportions of the background population. Black or African-American respondents interacted with a wider range of ages than other ethnic/racial groups, largely due to familial relationships. The egocentric networks of non-binary contacts had little age diversity. Black or African-American respondents in particular reported mixing with older or younger family members, which could increase the risk of transmission to vulnerable age groups. Understanding community mixing patterns can inform infectious disease risk, support analyses to predict epidemic size, or be used to design campaigns such as vaccination strategies so that community members who have vulnerable contacts are prioritized.

## Introduction

Human social networks tend to form as a set of dense, frequently homogeneous [1], clusters among people who share strong ties with each other, forming "communities," as they are called in network analysis. Weak ties between individuals then bridge these communities and

**Data Availability Statement:** A dataset for reproducibility and secondary analyses of these mixing patterns is made available through the Duke Research Data Repository (doi for the dataset and

documentation: https://doi.org/10.7924/r43f4zj2q).
Due to the sensitive nature of the questions asked
in this study and the higher risk of deductive
disclosure with relational data, survey respondents
were assured raw data would remain confidential
and would not be shared.

**Funding:** The project described was supported by
Grant/Cooperative Agreement Number
75D30120C09551 made to Duke University from
the Centers of Disease Control and Prevention
(CDC), US Department of Health and Human
Services (HHS), awarded to D.K.P., J.M., and K. B.-
E. D.K.P. and J.M. were also supported by the
National Institutes of Health (NIH) Eunice Kennedy
Shriver National Institute of Child Health and
Human Development (NICHD) (Grant Awards
R25HD079352 (awarded to J.M.), R21HD104431
(awarded to J.M. and D.K.P.), and R21HD101268
(awarded to J.M. and D.K.P)) and National Science
Foundation (Grant Award SES-2029790 (awarded
to J.M. and D.K.P)). The project was also
supported via consultations with the Duke Clinical
and Translational Science Institute, made possible
by NIH Award UL1TR002553. The contents are
solely the responsibility of the authors and do not
necessarily represent the official views of CDC, US
HHS, NIH, or NSF. The funders had no role in study
design, data collection and analysis, decision to
publish, or preparation of the manuscript.

**Competing interests:** The authors have declared
that no competing interests exist.

transmit infection (or information or ideas) in and out [2]. This characteristic of social networks has several implications for epidemic potential [3]. First, it can lead to bursty or episodic epidemics where small peaks in observed case counts result from increasing cases moving from group to group [4, 5]; cases increase first within one network community as the infection spins up between the strong ties then spreads out into a new community along the weak ties [6], which drives a new peak. This was observed during the SARS-CoV-2 pandemic as the early peaks formed within different social communities at different times despite being in the same city or geographic area [7, 8]. Second, it can result in a failure to notice an outbreak until it has established itself in a population [9], as with the HIV outbreak among injecting drug users in Indiana a few years ago [10, 11]. Since the infections were spreading among a group that was insular and did not access frequent care, the outbreak was out of control before it was noticed.

Community "social mixing patterns" form the basis for these transmission pathways of strong or weak ties [12, 13] and dictate transmission potential within and between communities defined by homophilous (preferential "like with like") ties. To better understand social mixing patterns and infection transmission potential in a small city (Durham County, North Carolina, USA), we enrolled diagnosed COVID-19 cases into a link tracing design study. In this study, we prioritized seeds (index cases) and incentivized peers (recruited contacts) who were under-represented in our cohort compared to the background population or who were a part of a sociodemographic group which was either typically under-represented in clinical research or experiencing an infection surge at the time of recruitment.

One aim of the study was to better understand how people mixed with each other in this sociodemographically diverse city [14], as the density of mixing within and between "communities" dictates both epidemic potential [15] and, importantly, is compounded by bias in case ascertainment given typical barriers to accessing care among ethnic or racial groups who have a historical mistrust of clinical care, who face language barriers, and/or who are socioeconomically disincentivized to seek testing or care (due to poor health coverage, inflexible work schedules, or lack of economic or social support during isolation or quarantine). We also wanted to use the chain referral methodology [16, 17] to assess the potential to leverage these social mixing patterns for recruitment and to increase engagement among groups who have been historically excluded from clinical research.

## Methods

### Cohort

We enrolled *seed* cases who were recently diagnosed with SARS-CoV-2 at a large, academic medical center (Duke University Health System) into a snowball sampling study in which they could recruit their local community contacts for respiratory infection sampling (ClinicalTrials. gov ID NCT04437706). Enrollment occurred from December 2020-July 2022 (19 months), as previously described [18]. Briefly, residents of Durham County, North Carolina, USA who were recently diagnosed with SARS-CoV-2 and aged 18 years or older were invited to complete a survey and then recruit their contacts to complete the survey and receive SARS-CoV-2 testing within a snowball sampling protocol to detect new SARS-CoV-2 infections in the community. Community contacts who chose to enroll were denoted as *peer* participants. All enrolled participants (seeds and peers) completed the same online survey, which collected self-reported sociodemographic information, SARS-CoV-2 infection and vaccination history, health history, health beliefs and behaviors, and asked about their close contacts ("cohabitants" and "non-cohabiting"). Respondents self-identified ethnicity, race, and gender were based on National Institutes of Health (USA) categories. Respondents answered questions about

their contacts' demographic traits (including age, ethnicity/race, and gender) and prior and current frequency of being in-person together; contacts could be any age, though enrollment as a seed or peer was restricted to people aged 18 years and older to reduce data collection from minors reporting on other minors. All participants who completed the survey had the opportunity to recruit peers for enrollment and sampling. Peers could be recruited even if they were not described as a contact in the survey. Informed consent was obtained from all participants within REDCap [19, 20] and accessed via a proprietary electronic platform [21]; trained study coordinators documented oral consent in cases where respondents needed assistance. This study was approved by the Duke University Health System (DUHS) Institutional Review Board (#Pro00105430). In accordance with the consent form, we cannot share raw data, but aggregated data are available to reproduce these findings (doi: https://doi.org/10.7924/r43f4zj2q).

## Egocentric networks and mixing patterns

Egocentric networks were created for each participant, composed of contacts described in the survey and recruited peers. The unit of analysis for the assessments of mixing was the paired participant and contact described.

In-group mixing on the categorical variables (ethnicity/race and gender) was calculated as a proportion of contacts with the same categorical selection as the respondent and we compared the mixing proportions to the proportions of the background population (Durham County, NC). Under the assumption of no preferential in-group mixing (no assortativity), the mixing proportions for each demographic group would follow the demographic proportions in the background population.

To check for similarity between egos and their contacts among a continuous variable, age, mixing was calculated as a Euclidean (average) distance between the respondent's age and the difference in reported ages of the contacts (Pearson's Phi statistic) [22]. Pearson's Phi compares ages across the entire egocentric network (not just from ego to contact) and accounts for number of contacts, so the measure does not increase simply for having a higher number of contacts. One Phi statistic is reported per ego; a lower score indicates more similarity within the ego network and a higher score indicates more dissimilarity. Upon aggregation, Pearson's Phi shows the distribution of age similarity between an ego and the alters (the contacts in the ego's network), and age mixing patterns were compared for each demographic trait. We compared in-group vs out-group ties among the respondents and their contacts described and peers referred (external-internal (E-I) index [22]) and compared the scores using Kruskal-Wallis tests for the sociodemographic traits collected.

For the enrolled peers, we constructed sender->receiver recruitment matrices of demographic traits comparing the peers who were enrolled into the study by the seed cases to look at recruitment patterns among the demographic traits of interest and assess the likelihood of recruitment among different groups. The matrices show the attribute of the referrer on the rows, with the attribute of the peer who was successfully recruited into the study and who completed the survey, as the column. The sum of the matrix is equal to the number of recruitment links, so study cohort members who recruited multiple people appear for each person they nominated who was successfully recruited into the study, consented, and completed a survey. Numbers along the diagonal (bolded) are in-group recruitment, where the referrer, or coupon distributor, successfully recruited someone with the same attribute value.

We conducted Chi-squared tests of associations between dichotomous groups (i.e., respondents 40 years or older vs younger than 40 years). All descriptive and statistical analyses were

conducted in R [23]. We used an alpha level of 0.05 as the threshold of significance for all analyses.

## Results

The 509 respondents (n = 384 seed cases and n = 125 peers) described 2,199 contacts (842 cohabitants and 1,357 other contacts) [Table 1], resulting in a total size across all egocentric networks of 2,544 people after de-duplicating across seeds, peers, and unenrolled contacts. Of the 125 peers recruited into the study, in-group recruitment was common: the majority were recruited by existing participants of the same ethnicity/race (Fig 1A diagonals). However, there was no in-group recruitment among Hispanic/Latinx participants; all Hispanic/Latinx peers were recruited by non-Hispanic participants (Fig 1A). Among the 6 peers recruited who identified as Hispanic, 3 (50%) either took the survey in Spanish or reported speaking Spanish primarily at home–two of these were recruited by a non-Hispanic Spanish-speaking household member and one was recruited by a non-Hispanic friend.

Recruitment tended to occur either within the same age band or one age band above or below. The sender-> receiver matrices show that in-group recruitment is successful among younger ages but that older age groups were recruited by a wider age range. Younger participants (less than 40 years) were frequently recruited as peers by existing participants of similar ages (Fig 1C column distribution) whereas peers aged 40 years and older were more likely to be recruited by an existing participant who was more than one age band away from the peer ($\chi^2(1) = 7.05$, $p<0.01$). Participants aged 20–29 years who managed to successfully recruit another participant mostly did so within their own age group, though they also recruited 30–39 and 50–59 year olds (Fig 1C row distribution). Participants aged 30–39 years recruited people from nearly every age band, though they were mostly recruited by their own age group and one band younger. In contrast, participants aged 50–59 years were recruited by people across the spectrum, from the group younger than 20 years all the way through 60–69 year olds, though they were not as effective at recruiting other participants. Across all age groups, most recruitments more than two age bands apart were between family members.

Among the 509 respondents who described contacts (N = 2,199), the egocentric network analysis revealed strong in-group mixing by ethnicity/race: 85% of contacts described by both White and Black respondents were the same race as the respondent. Homogeneity by ethnicity/race was somewhat lower for Asian (57% in-group) and Hispanic/Latinx (48% in-group) respondents, however Durham County (the target population) is composed of 5% Asian and 14% Hispanic/Latinx residents, so there was still a strong preference for homogeneous (in-group) ties (Fig 2). Overall, Asian, Black or African-American, Hispanic/Latinx, and White respondents demonstrated a significantly higher proportion of same-race contacts than would be expected had there been no preferential in-group mixing, based on the ethnic and racial proportions of the background population (Fig 2). There were significant differences in the proportion of in-group vs out-group ties by ethnicity/race (Kruskal Wallis $\chi^2(4) = 80.1$, $p<0.01$) [S1 Fig].

Respondents aged 40–69 years were most likely to mix with people of other ages (Fig 3), reporting family members such as children or parents and workplace contacts. Respondents aged 20–29 years had the most similarity in ages among their egocentric components, with dissimilarity increasing up to respondents aged 50–59 years. Compared to respondents of other ethnicities/races, Black or African-American respondents reported contacts who were significantly more dissimilar in age, having both higher median difference and a wider diversity by age when looking at each egocentric network component centered around a Black or African-American respondent (Fig 4). The contacts of Black or African-American respondents who

**Table 1. Sociodemographic characteristics of the background population (Durham County, North Carolina, USA) and cohort, as reported by the respondent.**

| | DURHAM COUNTY§ | Cohort Members with Completed Surveys | | Peers | | Total Contacts Described by Seed or Peer**,†† | |
|---|---|---|---|---|---|---|---|
| | (N = 321,488) | (N = 509) | | (n = 125) | | (N = 2,199) | |
| | (%) | n | (%) | n | (%) | n | (%) |
| **Gender** | | | | | | | |
| Male | 48 | 174 | (34) | 55 | (44) | 985 | (45) |
| Female | 52 | 326 | (64) | 65 | (52) | 1,188 | (54) |
| Non-binary / Other | | 9 | (2) | 5 | (4) | 26 | (1) |
| **Age, years*** | | | | | | | |
| 0–19 | 23 | 10 | (2) | 1 | (1) | 351 | (16) |
| 20–29 | 17 | 175 | (34) | 37 | (30) | 610 | (28) |
| 30–39 | 17 | 139 | (27) | 27 | (22) | 475 | (22) |
| 40–49 | 12 | 74 | (15) | 18 | (14) | 251 | (11) |
| 50–59 | 11 | 54 | (11) | 19 | (15) | 219 | (10) |
| 60–69 | 11 | 37 | (7) | 12 | (10) | 196 | (9) |
| 70–79 | 6 | 19 | (4) | 11 | (9) | 79 | (4) |
| 80+ | 3 | 1 | (<1) | 0 | (0) | 18 | (1) |
| **Ethnicity** | | | | | | | |
| Hispanic or Latino | 13 | 48 | (9) | 6 | (5) | 199 | (9) |
| Not Hispanic or Latino | 87 | 461 | (91) | 119 | (95) | 1,959 | (91) |
| unknown† | | 0 | | 0 | | 41 | |
| **Race** | | | | | | | |
| American Indian or Alaska Native | <1 | 0 | (0) | 0 | (0) | 4 | (<1) |
| Asian | 5 | 59 | (12) | 13 | (10) | 210 | (10) |
| Black or African-American | 36 | 105 | (21) | 29 | (23) | 408 | (19) |
| Native Hawaiian or Other Pacific Islander | <1 | 0 | (0) | 0 | (0) | 8 | (<1) |
| White | 42 | 294 | (58) | 74 | (59) | 1,312 | (61) |
| Other / 2+ races | 3 | 14 | (3) | 3 | (2) | 57 | (3) |
| None selected‡ | <1 | 37 | (7) | 6 | (5) | 159 | (7) |
| unknown† | | 0 | | 0 | | 41 | |
| **Primary Language¶** | | | | | | | |
| English | 83 | 451 | (89) | | | | |
| Spanish | 9 | 22 | (4) | | | | |
| Other | 8 | 36 | (7) | | | | |

Cohort members were enrolled participants with completed surveys, of which peer participants were a subset. All cohort members who completed a survey (N = 509) described their contacts (cohabitants and non-cohabitants; N = 2,199 total).

* Must be 18+ years to enroll

† Not included in proportion.

‡ Some enrolled participants (seeds and peers) or described contacts who were identified as Hispanic ethnicity declined to select a race.

¶ Primary Language here is language spoken at home for adults age 18+.

§ All data from Census Reporter, estimated as of 1-July-2019: https://censusreporter.org/profiles/05000US37063-durham-county-nc/ —site provides proportions, not numbers.

** Attributes of contacts are as reported by the respondent describing the contact, not reported by the contact or verified.

†† Some contacts may have been described as more than one respondent or may have enrolled in the study as a seed or peer participant; total size of ego networks is 2,544 unique people.

**ETHNICITY/RACE**

| | Coupon Recipients enrolled as Peers | | | | | |
|---|---|---|---|---|---|---|
| | Asian | Black or AA | Hispanic | White | 2 or more | |
| Asian | **8** | 1 | | 5 | | 14 |
| Black or AA | | **23** | | 9 | | 32 |
| Hispanic | 2 | 1 | | 3 | 1 | 7 |
| White | 3 | 4 | 5 | **57** | 1 | 70 |
| 2 or more | | | 1 | | **1** | 2 |
| | 13 | 29 | 6 | 74 | 3 | **125** |

**EDUCATION**

| | Coupon Recipients enrolled as Peers | | | | | |
|---|---|---|---|---|---|---|
| | Less than HS | HS or GED | Assoc / 2-yr | 4-year | Graduate | |
| Less than HS | | | | | | 0 |
| HS or GED | 1 | **6** | 3 | 2 | 2 | 14 |
| Assoc / 2-yr | | 2 | **1** | | | 3 |
| 4-year | 2 | 8 | 3 | **20** | 18 | 51 |
| Graduate | | 3 | 3 | 22 | **29** | 57 |
| | 3 | 19 | 10 | 44 | 49 | **125** |

**AGE (years)**

| | Coupon Recipients enrolled as Peers | | | | | | | |
|---|---|---|---|---|---|---|---|---|
| | 0-19 | 20-29 | 30-39 | 40-49 | 50-59 | 60-69 | 70-79 | 80+ |
| 0-19 | | | | | 2 | | | | 2 |
| 20-29 | | **27** | 10 | | 2 | | | | 39 |
| 30-39 | 1 | 6 | **14** | 4 | 2 | 4 | 3 | | 34 |
| 40-49 | | | 2 | **8** | 6 | 1 | 1 | | 18 |
| 50-59 | | | | 2 | **5** | | | | 7 |
| 60-69 | | 4 | 1 | 4 | 2 | **3** | 2 | | 16 |
| 70-79 | | | | | | 3 | **5** | | 8 |
| 80+ | | | | | | 1 | | | 1 |
| | 1 | 37 | 27 | 18 | 19 | 12 | 11 | 0 | **125** |

**GENDER**

| | Coupon Recipients enrolled as Peers | | | |
|---|---|---|---|---|
| | Female | Male | Non-binary | |
| Female | **46** | 29 | 1 | 76 |
| Male | 18 | **26** | | 44 |
| Non-binary | 1 | | **4** | 5 |
| | 65 | 55 | 5 | **125** |

**Fig 1. Sender->receiver recruitment matrices of the demographic traits of study participants who invited peers to participate (the coupon distributors) and the 125 peers who enrolled into the study (coupon recipients).** These matrices show the demographic traits of the existing study member ("coupon distributor") and the newly recruited study member ("coupon recipient") who chose to enroll in the trial (for each of the 125 peers). Coupon distributors who recruited multiple peers are reported multiple times in these matrices. The diagonals of the matrices represent coupon distributors who successfully recruited a peer who shares the same demographic trait. The ethnicity/race table shows that recruitment was most successful among members who identified as the same ethnicity-race. Most cross-generational recruitments in the age matrix were within families. Participants aged 20–29 years tended to distribute coupons within their own age group, to one age group older, and to 50–59 year olds. Participants aged 30–39, 40–49, and 60–69 years had a lot of diversity in coupon distribution.

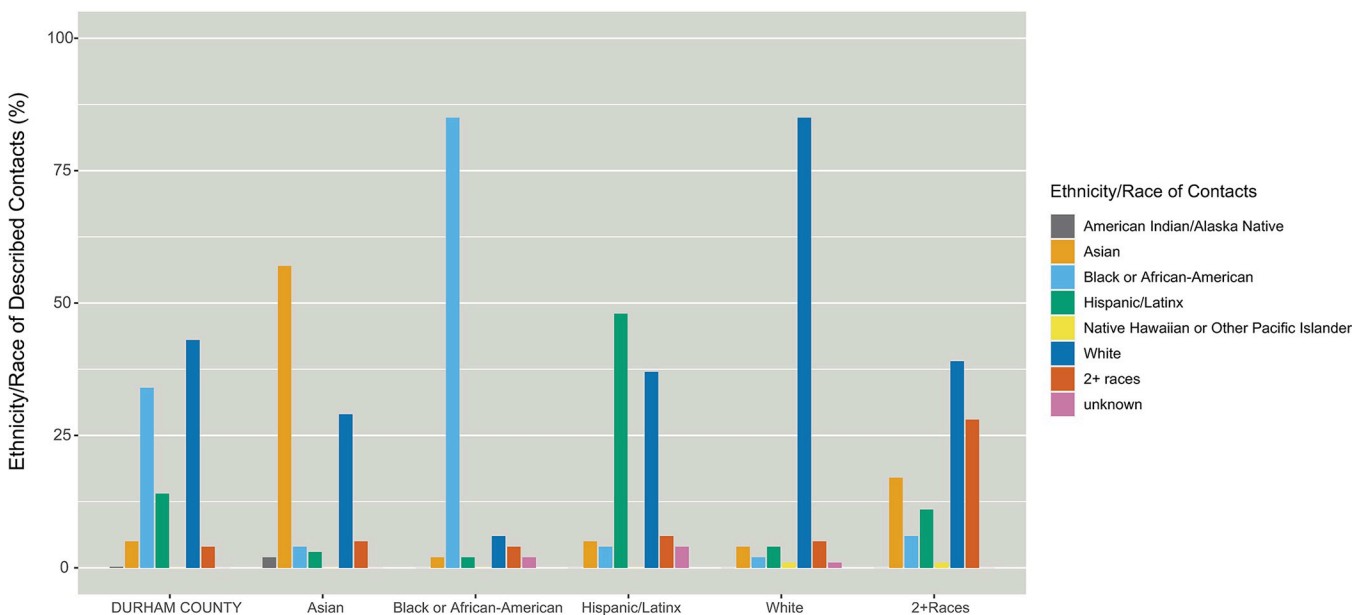

**Fig 2. Ethnic/racial proportion of target population (Durham County, NC) compared to ethnic/racial proportion of contacts per ethnicity/race of respondent.** Durham County, NC, is a racially diverse area where more than half of residents are non-White. Respondents are split by self-reported ethnicity/race on the x-axis, with the ethnic/racial proportions of their contacts (as reported by the respondent) displayed on the y-axis. Asian, Black or African-American, Hispanic/Latinx, and White respondents had a significantly higher proportion of same-race (in-group) contacts than would be expected under the assumption of no assortativity based on the ethnic and racial proportions of the background population.

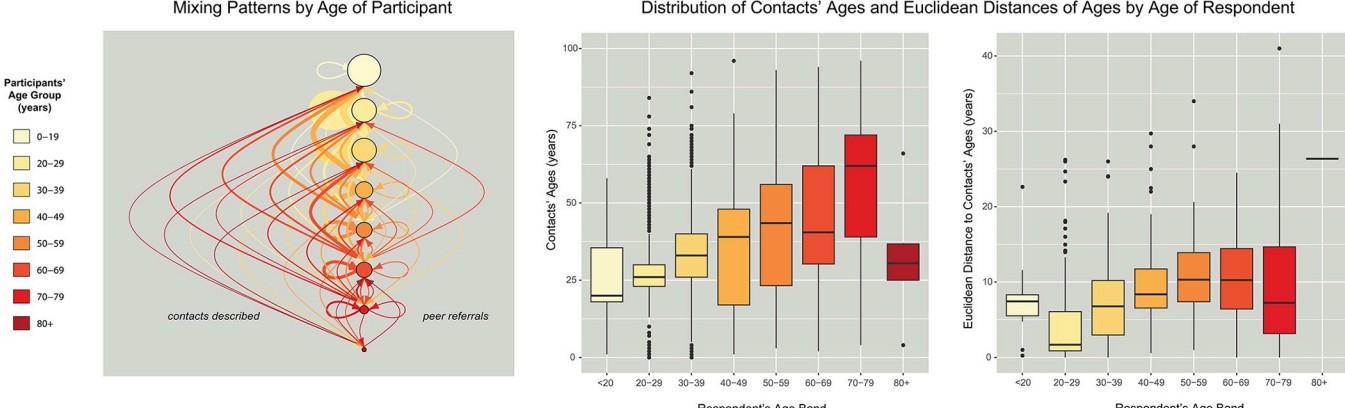

**Fig 3. Age-group mixing among the study cohort members.** LEFT: Graphic showing the age of enrolled participant (circles colored by age band) and the contacts described on the left and the successful peer recruitments on the right. Size of circle corresponds to group size of enrolled participants. Arrow color indicates the age group from which the contact description or peer referral originated and arrow width corresponds to the number of contacts or peer recruitments of the other group; as illustrated, many contacts and referrals were within the same age band. CENTER: Box plot of reported age of contacts by 10-year age band of the respondent. A general trend is seen where the age of contacts increases as the age of the respondent increases, with exceptions for respondents aged younger than 20 years or aged 80 years or older. RIGHT: Box plot of the dispersion of contacts' ages around the respondent's age. Age mixing was calculated as a Euclidean distance between the age of the respondent and the reported contacts. A smaller value on the y-axis indicates that contacts are typically closer in age to the respondent. A general u-shaped trend is seen where respondents aged 40–69 years have the greatest overall difference in ages between the respondent and their contacts, as respondents in these 10-year bands were more likely to report older or younger family members as contacts; for this reason, respondents aged 80 years and older had the greatest difference between their own ages and the typical ages of their contacts (as shown by the highest y-values) as they tended to report younger family members as contacts.

were more than one age band apart were more likely to be familial ties compared to other ethnic/racial groups. Overall, non-binary respondents were more likely to mix with people of similar age to themselves, but there were no significant differences in age mixing patterns between males and females (Fig 5).

Affiliates of the university, which included students, staff, and faculty, were more likely to mix with people of similar ages (within 1 age band) than people who were not affiliated with the university ($\chi^2(1) = 4.50$, $p = 0.03$), but were equally likely to report contacts of the same ethnicity/race ($\chi^2(1) = 0.67$, $p = 0.41$).

## Discussion

Though Durham County, NC, is ethnically and racially diverse [14], we observed a high degree of ethnic/racial homogeneity (homophily) among contacts described by respondents. Age mixing differed by ethnicity/race and age group, with Black or African-American respondents more likely to have alters of other age groups and respondents of younger ages more likely to have alters of similar ages. The larger age dissimilarity among the egocentric networks of 50–59 year-old respondents is not surprising considering that people in this age group in the United States are frequently in the work force and/or are caring for people in different generations.

Social mixing patterns have implications for the potential of an infection to move through a community. First, the smaller minority groups were disproportionately more likely to have contacts with other ethnic/racial groups, increasing the potential to bring infection into the group. Second, the ethnic/racial homophily means that an infection that reaches an ethnic or racial group is likely to become highly prevalent within that group, which can have knock-on effects during an outbreak or epidemic if the community's resources or capacity needed to care for the ill people in that group become stretched. Third, the age mixing patterns suggest that people who were not believed to be at high risk of poor outcomes during the early phases

## Distribution of Euclidean Distances of Ages by Ethnicity and Race of Respondent

**Fig 4. Age-group mixing between the respondent and reported contacts by demographic trait of the respondent.** These figures show the distribution of respondent age, contact age, and Euclidean distances between the age of the respondent and the reported contacts in the respondent's egocentric network, by the ethnicities/races of the respondents. LEFT: Box plot of respondent's age in years by respondent's self-reported ethnicity/race. Black or African-American and White respondents tended to be slightly older and across a larger span of ages than respondents who identified as Hispanic/Latinx, Asian, or multiple races. CENTER: Box plot of reported age of contacts by the respondent's self-reported ethnicity/race. The contacts of Black or African-American and White respondents tended to span more ages. RIGHT: Box plot of the dispersion of contacts' ages around the respondent's age. Age mixing was calculated as a Euclidean distance between the age of the respondent and the reported contacts. A smaller value on the y-axis indicates that contacts are typically closer in age to the respondent. Asian respondents and those who identified with two or more races tended to have contacts closer in ages to themselves (as shown by the y-values). Although Black or African-American and White respondents appear similar in aggregate when looking at the ages of the respondents and the ages of the respondents' contacts, the Euclidean distance (typical difference in ages between respondent and contact) is higher for Black or African-American than White contacts, reflecting a higher likelihood to report contacts who are not similar in age to the respondent.

of the SARS-CoV-2 pandemic when mitigation strategies were being developed were in fact mixing with very vulnerable groups. These strategies included an age-tiered vaccination strategy [24] and even some discussions about age-tiered return-to-work plans.

Asian and Hispanic/Latinx respondents reported a high degree of in-group mixing, yet still had a substantial proportion of out-group contacts, most commonly White, described in the social network survey. Though more than half of Durham County residents are non-White, Asian and Hispanic/Latinx residents are a relatively smaller proportion of the population. This can lead to a higher likelihood of bridging ties to other groups simply due to probability [25], which provide pathways for an infection to enter a community. The risk of infection transmission is then compounded in smaller groups which are segregated by language or culture: they often tend toward higher cohesion as there are fewer options for ties [25], which leads to a denser set of in-group network ties and in turn increases the risk for in-group transmission if an infection enters the community.

Black and African-American respondents in this cohort had higher racial insularity among close contacts and thus fewer opportunities for an infection to enter the community, but mixing patterns put vulnerable age groups at risk. Most Black or African-American respondents were 30–51 years (IQR; Fig 4), which was not a priority group for SARS-CoV-2 vaccination based on age. However, Black or African-American respondents were more likely to report

Typical Distances between Respondent's and Contact(s)' Ages

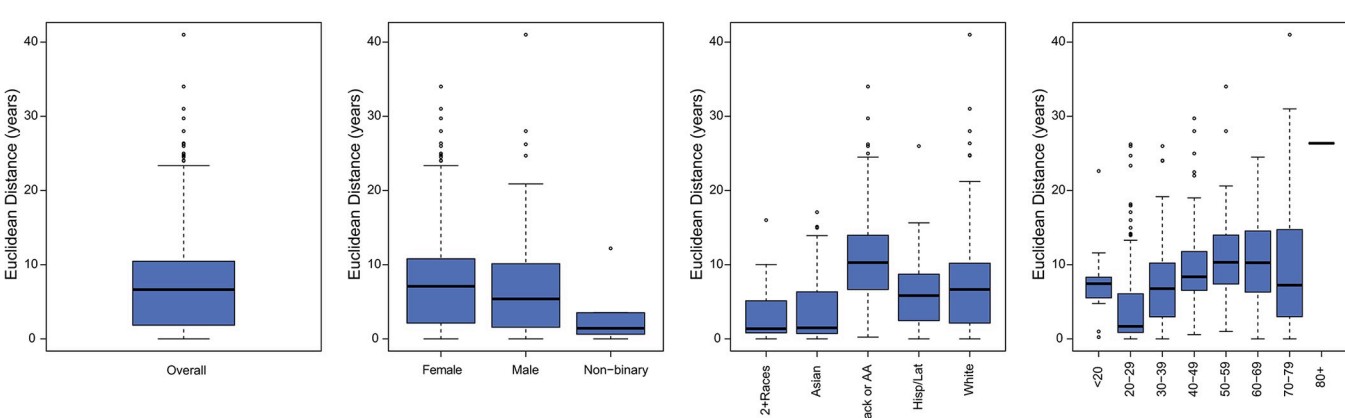

**Fig 5. Age-group mixing between the respondent and reported contacts by demographic trait of the respondent.** Age mixing was calculated as a Euclidean distance between the age of the respondent and the reported contacts in the respondent's egocentric network. A smaller value on the y-axis indicates that contacts are typically closer in age to the respondent. A) Overall Euclidean distance of typical age difference between the respondent and the respondent's reported contacts for all cohort members. B) Euclidean age distance by self-reported gender; non-binary respondents tended to have contacts similar to their own age than the overall group. C) Euclidean age distance by self-reported ethnicity/race; Black or African-American respondents were most likely to report contacts with dissimilar ages whereas Asian respondents and those reporting 2 or more races were more likely to have contacts closer in age to themselves. This repeats the right-hand panel in Fig 4 to show the trend compared to other demographic traits. D) Euclidean age distance by age; this repeats the right-hand panel in Fig 3 to show the trend compared to other demographic traits.

significant close contact with people who were in vulnerable age groups and who were either not eligible for vaccination for quite some time (those aged less than 18 years) or who were older and at higher risk of waning immunity given age-related immune senescence [26]. Though older age groups had higher vaccination coverage than middle-aged groups among all ethnicities/races in April 2021 [27], it was still below the targets to convey maximum protection. As the goal of the tiered vaccination strategy was to reduce overall mortality in the community, understanding mixing patterns could be used to inform roll-out strategies such that the priority groups for vaccine distribution are identified with the goal of reducing inequity in morbidity and mortality outcomes.

These mixing patterns can also be used to design programs to provide social or economic support for groups who are more likely to mix with vulnerable groups, to reduce the risk of transmission and thus poorer outcomes [28], before an effective vaccination or treatment is available. In the context of limited resources, understanding community mixing patterns can increase efficiency in deployment and maximize the effects of allocating scarce resources. This approach builds equity and reduces health gaps by ethnicity/race or socioeconomic status by accurately reflecting the potential flow of infections, but also disparities in the flow of resources throughout a community as it affects more insular or less resourced groups. Understanding the overall community flow between groups permits targeted advertising as well as development of proactive strategies to intervene among highly connected groups for whom a targeted intervention protects not only that group but also the connected communities. Possibilities for future work testing the applicability of these findings includes testing how quickly this information can be collected if it is needed for public health response, testing whether this information can be proactively collected and maintained for these purposes, or testing concordance with other types of information that can approximate mixing patterns, such as cell phone or transportation data.

There are some limitations related to the interpretation of the age mixing patterns. First, we used an online survey instrument which may have been more attractive to younger age groups

that are more comfortable with the technology. Second, students were over-represented in the cohort compared to the background population. This may have increased the in-group age mixing trend observed among younger study participants. Other limitations include the cohort's demographic representation of the background population. Younger participants, women, Asians, and White participants were over-represented in the cohort and Black or African-American, Hispanic/Latinx participants, and primarily Spanish-speaking participants were under-represented. Therefore, mixing patterns may have differed from what we are reporting among the under-represented groups had there been higher enrollment. However, even university affiliates, including faculty, staff, and students, reported significant in-group mixing on ethnicity/race despite being a part of a diverse academic setting, which was operating in-person during the study, and having classmates or colleagues of all ethnicities and races, so these patterns may be robust even without mixing being forced as a result of ethnic/racial homogeneity in the setting or endogenous effects such as triadic closure [29, 30] promoting same-race friendships as a result of familial or generational community relationships. High socioeconomic status is associated with larger and more varied networks [31]. Since people with high educational attainment were overrepresented across demographic categories in our sample, and we still found significant in-group mixing, we would expect to find similar if not stronger patterns in the under-represented groups.

A major strength of this analysis is the ability to compare the contacts that participants chose to describe with those who were successfully recruited. Peer recruitment and in-group mixing paint different pictures of mixing among Black or African-American respondents in particular. While 85% of the contacts described by Black or African-American respondents were also Black or African-American (Fig 2), only 72% (23/32) of the peers recruited by a Black or African-American respondent were also Black or African-American (the other 9 were all non-Hispanic White).

It is possible that these mixing patterns could be positively leveraged to engage historically excluded groups in clinical care or clinical research [32]. Among this cohort, Black or African-American participants were able to successfully recruit within race, which could have been a way to increase participation among a group that has been historically excluded from research. Leveraging trust based on relationship may be another strategy; in this study, many participants were recruited along familial ties. Older participants in particular appeared amenable to recruitment by existing study participants, who in many cases were family members. These types of ties may also be vitally important when seeking to recruit cohort members via link tracing who are not part of the workforce or other social networks. When designing a chain referral strategy, selecting seeds whose value lies in the "second order" [33] or local network rather than immediate, first-degree network may be a way to reach participants who experience mistrust or have similar barriers to engagement.

This cohort provided valuable information about the ethnic and racial insularity of a highly diverse population, which is important when considering epidemic preparedness. The smaller minority groups were less insular, which could increase risk of transmission into the group. Assessment of community mixing patterns can also provide additional insights into transmission or transmission risk across the entire location in two ways. First, it provides clues as to what constitutes a network community in the local area and the properties of the weak tie bridges between the communities. This understanding can differentiate incidence resulting from continued transmission vs peaks within each micro-population or community, and understanding the weak ties between communities provides a sense of how an infection might travel through a community for the purposes of intervening to slow transmission in the overall community and/or forecasting resources. Second, heterophilous (out-group) ties can increase risk in one group by being linked to a higher prevalence group [34]. This phenomenon has been

observed when having older partners who have had a longer time in which to acquire HIV leads to high risk among younger men who have sex with men in San Francisco [35] and young women in South Africa [36].

These data and findings add to the literature describing empirical social mixing patterns [37] and can be used to inform simulations or epidemic models which aim to predict epidemic spread, as preferential vs random mixing affect epidemic trajectories [38]. Here, we have observed the hypothesized modularity in a real community based along ethnic/racial lines, though with different potential within each community for spread to more vulnerable groups. The combination of multiple dimensions of sociodemographic information which may affect diffusion–in this case, ethnicity/race + age–has the potential to strengthen epidemic models. In the US, socioeconomic disparities frequently follow ethnic/racial lines. Incorporating these mixing patterns, including those where cross-familial ties indicate caregiving responsibilities, adds elements of realism and equity to models, both of which are key parameters when developing interventions [39].

For an ongoing or longer-term epidemic, identifying the social network communities, the groups of people with dense sets of strong ties, is a key part of reducing onward transmission. First, incorrectly aggregating groups which appear to have similar traits or risk factors may obscure the actual overall risk to smaller network communities that have circulating uncontrolled infection and thus higher prevalence and incidence. For example, Latinos in North Carolina are frequently grouped together for HIV prevention efforts, though foreign-born Latinos with HIV tended to mix with other foreign-born Latinos and be part of heterosexual networks whereas US-born Latinos with HIV tended to mix with other men who have sex with men and were less likely to be in a putative transmission cluster with another Latino [40]. Incorrectly aggregating groups may lead to delay in identifying a problem if prevalence seems lower because the overall prevalence is diluted when groups are combined despite not having equal risk. Second, failing to identify which sub-groups have the highest prevalence (and thus risk of new infection for group members) can mean that an intervention is applied in the wrong community, as with HIV pre-exposure prophylaxis where the people most at risk may not have access, or as with the earliest SARS-CoV-2 testing sites frequently being set up near communities at lower risk due to a sadly common trend of tilting toward more resourced communities [41], which additionally tended to contain residents who had the ability to work and learn from home. Finally, aggregating a group of people based on a shared demographic trait rather than seeking to understand the relationships along which infections move can lead to failing to elucidate the transmission pathways.

These mixing patterns may also be a tool to increase participation in preventive care or research. Network interventions have shown success in promoting and diffusing positive health behaviors [42–45]. Among historically excluded groups, who often have a valid mistrust of research and medical institutions [46], selecting seed cases who may not be a part of a historically excluded group but have a propensity to recruit a member from this group, along a trusted social tie, may be a way to increase engagement. Understanding and utilizing social mixing patterns has multiple benefits related to health processes that diffuse through a network, which can include promoting behaviors [47–49], clinical research recruitment or engagement [44], or slowing infectious transmission [50].

## Supporting information

**S1 Fig. In-group vs out-group ties by ethnicity/race of respondents.** Homophily scores by ethnicity/race, as measured by external-internal (E-I) index, among enrolled seeds (n = 376) and peers (n = 116) who described or recruited at least one person (of the cohort N = 509, 8

seeds and 9 peers did not describe any contacts or recruit anyone else into the study). The unit of analysis for the E-I index is the egocentric network. A score of +1 indicates all out-group ties and a score of -1 indicates all in-group ties. A Kruskal-Wallis test identified significant differences between the groups ($\chi^2(4) = 80.1$, $p<0.01$). A pairwise Dunn test between groups with a Bonferroni correction identified statistically significant differences between all pairs except Black or African-American respondents and White respondents and between Hispanic or Latinx respondents and those who identified as 2 or more ethnicities/races.
(TIF)

## Acknowledgments

We would like to acknowledge the contributions of Brisa Barajas-Gomez, Aleah Bowie, Victoria Christian, Jessilyn Dunn, Jordan Hairston, Erich Huang, Jonathan McCall, Micah McClain, Yuhan McGee, Margaret Pendzich, Elizabeth Petzold, Karnika Singh, Maria Luisa Solis-Guzman, Chris Woods, and Mark Yacoub to the work described here. The authors thank Duke AI Health for support provided.

## Author Contributions

**Conceptualization:** Dana K. Pasquale, Keisha L. Bentley-Edwards, Andrew Olson, James Moody.

**Data curation:** Whitney Welsh, Andrew Olson.

**Formal analysis:** Dana K. Pasquale, Whitney Welsh.

**Funding acquisition:** Dana K. Pasquale, Keisha L. Bentley-Edwards, Andrew Olson, James Moody.

**Investigation:** Dana K. Pasquale, Whitney Welsh, Keisha L. Bentley-Edwards, James Moody.

**Methodology:** Dana K. Pasquale, Andrew Olson, James Moody.

**Project administration:** Andrew Olson.

**Supervision:** Dana K. Pasquale, Whitney Welsh, Andrew Olson.

**Validation:** Whitney Welsh, Andrew Olson.

**Visualization:** Dana K. Pasquale, Whitney Welsh.

**Writing – original draft:** Dana K. Pasquale.

**Writing – review & editing:** Dana K. Pasquale, Whitney Welsh, Keisha L. Bentley-Edwards, Andrew Olson, Madelynn C. Wellons, James Moody.

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
