## [Decision Letter · Decision Letter 0]

31 Jan 2024

PONE-D-23-41812Social Mixing Patterns Lead to Insights about Infectious Transmission PathwaysPLOS ONE

Dear Dr. Pasquale,

Thank you for submitting your manuscript to PLOS ONE. After careful consideration, we feel that it has merit but does not fully meet PLOS ONE’s publication criteria as it currently stands. Therefore, we invite you to submit a revised version of the manuscript that addresses the points raised during the review process.

I took at a careful look at the reviewer's comments and it appears that the revised manuscript is much improved. Please address the remaining concerns and also ensure that all data underlying the findings is made publicly available, prior to resubmission (see The PLOS Data policy). Please include with your resubmission a point-by-point reply to the reviewer's comments.

We look forward to receiving your revised manuscript.

Kind regards,

Claus Kadelka

Academic Editor

PLOS ONE

Journal Requirements:

Reviewers' comments:

Reviewer's Responses to Questions

**Comments to the Author**

1. Is the manuscript technically sound, and do the data support the conclusions?

Reviewer #1: Yes

Reviewer #2: Partly

2. Has the statistical analysis been performed appropriately and rigorously? 

Reviewer #1: Yes

Reviewer #2: I Don't Know

3. Have the authors made all data underlying the findings in their manuscript fully available?

Reviewer #1: No

Reviewer #2: No

4. Is the manuscript presented in an intelligible fashion and written in standard English?

Reviewer #1: Yes

Reviewer #2: Yes

5. Review Comments to the Author

Reviewer #1: The study describes the results of a social contact survey that was conducted over 19 months between 2020 and 2021 in a small city in North Carolina, USA.

The authors recruited 509 adults by using a snowball sampling design to enroll people recently diagnosed with SARS-CoV-2. Each participant was asked to describe their contacts.

Results of the study show high homophily between ethnic/racial groups. Also, age assortativity was lower among Black or African-American compared to other groups.

Characterizing social contacts among ethnic/racial groups is important to improve the representativity of epidemic models, and the study represents a useful addition to the literature. Overall, the study does not present significant shortcomings and I am positively inclined toward acceptance.

I have a few comments, mainly related to the methods, that I hope the authors could address to improve the manuscript.

1. The title is quite generic and presents a fact that is not new (the connection between social mixing and epidemic spreading). Moreover, the title emphasizes the epidemiological relevance of the study, however, the study itself does not contain any concrete epidemiological analysis (either from infection data or from simulations).

I would suggest changing the title to better reflect the actual content of the study which is more focused on the ethnic/racial mixing in a small community.

2. One main limitation of the study is that the cohort is not representative of the county population, therefore a comparison with the demographics of the whole county does not seem the best way to expose the difference in assortative behaviour from the background population (Figure 2). I would rather compare the assortativity of the sample with a random mixing assumption, where the contacts between groups are computed under the assumption that the cohort mixed randomly. Would that make sense?

3. To my knowledge, measuring the Euclidean distance between ages is not a standard way to compute similarity of contacts. I would suggest adding an equation to better explain how this is computed and provide a reference. It is not clear to me why do you need a distance metric to compute difference between age bands. Counting the difference in the number of age bands between individuals would work equally well. Euclidean distance is also quite difficult to interpret.

4. Data availability. I understand the raw data cannot be shared. However, could some aggregated form of contact matrices be shared?

5. Minor comment. I think the main value of the study lies in the characterization of social mixing along the racial/ethnic dimension. This is relevant to improve epidemic models that usually lack these components. I would suggest emphasizing this aspect in the Discussion. A recent study (Ma, K. C., Menkir, T. F., Kissler, S., Grad, Y. H., & Lipsitch, M. (2021). Modeling the impact of racial and ethnic disparities on COVID-19 epidemic dynamics. Elife, 10, e66601.) has included racial mixing in epidemic modelling, solely based on census data. In general, addressing the socio-economic disparities in modelling is key to advance the field (see for instance, Tizzoni, M., Nsoesie, E. O., Gauvin, L., Karsai, M., Perra, N., & Bansal, S. (2022). Addressing the socioeconomic divide in computational modeling for infectious diseases. Nature Communications, 13(1), 2897).

Reviewer #2: Overall, the manuscript engages in an interesting conversation, the nexus between social mixing patterns and infection transmission and promises to project some insightful policy and theoretical implications. However, I think some further clarifications of the following issues may help enhance its potential contribution to policy and theory. I have attached the reviewer comments in a separate file. The edictor may share the file with the authors.

6. PLOS authors have the option to publish the peer review history of their article (what does this mean?). If published, this will include your full peer review and any attached files.

Reviewer #1: **Yes: **Michele Tizzoni

Reviewer #2: No

---

## [Author Response · Author response to Decision Letter 0]

12 Apr 2024

Dear editorial team,

We are grateful for the reviewers’ time and thoughtful review. We have made the suggested revisions as described below, and appreciate the comments as we believe that these changes have strengthened the manuscript.

We indicate insertions to text below with underlines and deletions with strike-through.

We are incredibly appreciative of the reviewers’ suggestions to make the study and outcomes more clear. We believe that the changes and clarifications suggested by the reviewers improve the manuscript and request to submit this revision with increased length.

Thank you.

Comments to the Author

1. Is the manuscript technically sound, and do the data support the conclusions?

Reviewer #1: Yes

Reviewer #2: Partly

2. Has the statistical analysis been performed appropriately and rigorously? 

Reviewer #1: Yes

Reviewer #2: I Don't Know

3. Have the authors made all data underlying the findings in their manuscript fully available?

Reviewer #1: No

Reviewer #2: No

Response from co-authors: please note that we have developed a dataset for sharing; please note these details in the Data Disclosure statement.

4. Is the manuscript presented in an intelligible fashion and written in standard English?

Reviewer #1: Yes

Reviewer #2: Yes

5. Review Comments to the Author

Reviewer #1: The study describes the results of a social contact survey that was conducted over 19 months between 2020 and 2021 in a small city in North Carolina, USA.

The authors recruited 509 adults by using a snowball sampling design to enroll people recently diagnosed with SARS-CoV-2. Each participant was asked to describe their contacts.

Results of the study show high homophily between ethnic/racial groups. Also, age assortativity was lower among Black or African-American compared to other groups.

Characterizing social contacts among ethnic/racial groups is important to improve the representativity of epidemic models, and the study represents a useful addition to the literature. Overall, the study does not present significant shortcomings and I am positively inclined toward acceptance.

I have a few comments, mainly related to the methods, that I hope the authors could address to improve the manuscript.

1. The title is quite generic and presents a fact that is not new (the connection between social mixing and epidemic spreading). Moreover, the title emphasizes the epidemiological relevance of the study, however, the study itself does not contain any concrete epidemiological analysis (either from infection data or from simulations).

I would suggest changing the title to better reflect the actual content of the study which is more focused on the ethnic/racial mixing in a small community.

Response: Thank you for this suggestion. We have changed the title.

2. One main limitation of the study is that the cohort is not representative of the county population, therefore a comparison with the demographics of the whole county does not seem the best way to expose the difference in assortative behaviour from the background population (Figure 2). I would rather compare the assortativity of the sample with a random mixing assumption, where the contacts between groups are computed under the assumption that the cohort mixed randomly. Would that make sense?

Response: We agree with the reviewer’s criticism of the representativeness of the cohort and very much appreciate the suggestion. As these are ego networks, we do not know the context of the larger social network for the respondents. Per Perry, Pescosolido, & Barry section 10.2.3 “Non-Choices” (cited in the manuscript), we therefore must select the best estimate for the possibility of alter choices which was the County in our case. 

In response to this suggestion, we have developed a Supplemental Analysis where we calculate the E-I index for the key demographic traits that we collected. If we have not interpreted the suggested analysis as it was intended then we are happy to make another revision if the editor allows it. 

3. To my knowledge, measuring the Euclidean distance between ages is not a standard way to compute similarity of contacts. I would suggest adding an equation to better explain how this is computed and provide a reference. It is not clear to me why do you need a distance metric to compute difference between age bands. Counting the difference in the number of age bands between individuals would work equally well. Euclidean distance is also quite difficult to interpret.

Response: We agree that this is not the usual metric. However, we wanted to retain the granularity as much as possible. Durham County, NC, has several universities and a large academic medical center. By using ages in years rather than age bands we are able to distinguish relationships that might be conflated otherwise, such as mixing between a 29 and a 30 year old (more or less a same-age contact) vs a 20 and a 39 year old (not a same-age contact). We have cited the source (Perry, Pescosolido, and Borgatti, 2018).

4. Data availability. I understand the raw data cannot be shared. However, could some aggregated form of contact matrices be shared?

Response: We have provided an open-source dataset for reproducibility and secondary analysis of the mixing patterns presented (doi: 10.7924/r43f4zj2q). The details are in the manuscript. 

5. Minor comment. I think the main value of the study lies in the characterization of social mixing along the racial/ethnic dimension. This is relevant to improve epidemic models that usually lack these components. I would suggest emphasizing this aspect in the Discussion. A recent study (Ma, K. C., Menkir, T. F., Kissler, S., Grad, Y. H., & Lipsitch, M. (2021). Modeling the impact of racial and ethnic disparities on COVID-19 epidemic dynamics. Elife, 10, e66601.) has included racial mixing in epidemic modelling, solely based on census data. In general, addressing the socio-economic disparities in modelling is key to advance the field (see for instance, Tizzoni, M., Nsoesie, E. O., Gauvin, L., Karsai, M., Perra, N., & Bansal, S. (2022). Addressing the socioeconomic divide in computational modeling for infectious diseases. Nature Communications, 13(1), 2897).

Response: Thank you very much for this suggestion. We have included a new paragraph in the Discussion addressing this very important point, referencing both of these articles.

Reviewer #2: Overall, the manuscript engages in an interesting conversation, the nexus between social mixing patterns and infection transmission and promises to project some insightful policy and theoretical implications. However, I think some further clarifications of the following issues may help enhance its potential contribution to policy and theory. I have attached the reviewer comments in a separate file. The editor may share the file with the authors.

PONE-D-23-41812

Social Mixing Patterns Lead to Insights about Infectious Transmission Pathways

PLOS ONE 

Overall impression

The manuscript engages in an interesting conversation, the nexus between social mixing patterns and infection transmission and promises to project some insightful policy and theoretical implications. However, I think some further clarifications of the following issues may help enhance its potential contribution to policy and theory.

Introduction 

1. I believe that the introduction of a research paper needs to present a conversation-problem-solution (contribution) picture of the study.

1) Conversation: what is the scholarly literature that the paper is joining?

2) Problem: what issue in that literature is the paper trying to solve?

3) Solution (contribution): how exactly is the paper addressing that problem?

I see that the authors have projected the conversation they are joining, i.e., the implications of social mixing patterns on infectious (SARS-CoV-2) transmission pathways. They have also indicated in lines 92-94 that they are joing the conversation ‘To better understand social mixing patterns and infection transmission potential in a small city (Durham County, North Carolina, USA)’. Certainly, understanding social mixing patterns and infection transmission is important. However, I do not see an established basis for this aim. Is it a knowledge gap to warrant the study? Can the authors point to what is missing in the current literature on the nexus between social mixing patterns and infection transmission? 

Response: Simulation studies show that network structure and modularity affect diffusion, in this case, epidemic spread and trajectory. In this study, we observed the hypothesized modularity in a real community based on ethnicity and race, though there are differences by age. We have noted these implications and cited recent works related to diffusion among communities in the Discussion.

Methods

2. I am not familiar with the authors’ analytic lens, i.e., egocentric network analysis. However, I think some further explanation of methodological choices can help readers. For instance, why was the study limited to people aged 18 years and older? Why limit the study to only people newly diagnosed with SARS-CoV-2? What does newly diagnosed mean?

Response: Thank you for this comment. We have amended the “Cohort” paragraph of METHODS to include these details, including that we restricted enrolment to 18 and older because we wanted to reduce the data collection from minors who would be more likely to describe other minors who had not consented to participate. 

3. I was also wondering how the categorical variables (ethnicity/race and gender) in their online survey were defined? Were they researcher or participant defined? If these were researcher-defined, what standard categorization was used? And why? I believe this is fundamental as there are different and evolving categorizations of these variables.

Response: Sociodemographic categories were provided in the survey based on National Institutes of Health categories and respondents were asked how they identify. We have amended the “Cohort” paragraph of METHODS to include this information.

Results 

I have no expertise in the kind of analysis presented to enable me provide a useful review comment.

Discussion 

4. The authors have provided a detailed discussion of their findings in relation to their research objectives. They have also situated their findings within existing literature. I also see some attempt to point out the strengths and limiations of the study. However, I think a some clarity is needed to point out the concrete policy and theoretical implications of the results. What do policy-makers and health organisations do to manage infection transmission pathways more effectively? And what are new research questions (further research directions) emerge from the findings? 

Response: Thank you for this comment. We have added text to the Discussion addressing these very important questions (underlined text is new):

These mixing patterns can also be used to design programs to provide social or economic support for groups who are more likely to mix with vulnerable groups, to reduce the risk of transmission and thus poorer outcomes28, before an effective vaccination or treatment is available. In the context of limited resources, understanding community mixing patterns can increase efficiency in deployment and maximize the effects of allocating scarce resources. This approach builds equity and reduces health gaps by ethnicity/race or socioeconomic status by accurately reflecting the potential flow of infections, but also disparities in the flow of resources throughout a community as it affects more insular or less resourced groups. Understanding the overall community flow between groups permits targeted advertising as well as development of proactive strategies to intervene among highly connected groups for whom a targeted intervention protects not only that group but also the connected communities. Possibilities for future work testing the applicability of these findings includes testing how quickly this information can be collected if it is needed for public health response, testing whether this information can be proactively collected and maintained for these purposes, or testing concordance with other types of information that can approximate mixing patterns, such as cell phone or transportation data.

6. PLOS authors have the option to publish the peer review history of their article (what does this mean?). If published, this will include your full peer review and any attached files.

Do you want your identity to be public for this peer review? For information about this choice, including consent withdrawal, please see our Privacy Policy.

Reviewer #1: Yes: Michele Tizzoni

Reviewer #2: No

We would also like to thank the editor for the notes / suggestions in the email:

 Response: We have corrected the file names and main body formatting.

 Response: We have moved the information from the ethics statement on the title page to Methods.

Response: We have checked all of the references and do not find any that have been retracted. However, if we failed to identify one that has been then we will select another citation that has not been retracted.

---

## [Decision Letter · Decision Letter 1]

30 Apr 2024

Homophily and social mixing in a small community:  Implications for infectious disease transmission

PONE-D-23-41812R1

Dear Dr. Pasquale,

We’re pleased to inform you that your manuscript has been judged scientifically suitable for publication and will be formally accepted for publication once it meets all outstanding technical requirements.

Kind regards,

Claus Kadelka

Academic Editor

PLOS ONE

Additional Editor Comments (optional):

Reviewers' comments:

Reviewer's Responses to Questions

**Comments to the Author**

1. If the authors have adequately addressed your comments raised in a previous round of review and you feel that this manuscript is now acceptable for publication, you may indicate that here to bypass the “Comments to the Author” section, enter your conflict of interest statement in the “Confidential to Editor” section, and submit your "Accept" recommendation.

Reviewer #1: All comments have been addressed

Reviewer #2: All comments have been addressed

2. Is the manuscript technically sound, and do the data support the conclusions?

Reviewer #1: Yes

Reviewer #2: Yes

3. Has the statistical analysis been performed appropriately and rigorously? 

Reviewer #1: Yes

Reviewer #2: I Don't Know

4. Have the authors made all data underlying the findings in their manuscript fully available?

Reviewer #1: Yes

Reviewer #2: No

5. Is the manuscript presented in an intelligible fashion and written in standard English?

Reviewer #1: Yes

Reviewer #2: Yes

6. Review Comments to the Author

Reviewer #1: (No Response)

Reviewer #2: (No Response)

7. PLOS authors have the option to publish the peer review history of their article (what does this mean?). If published, this will include your full peer review and any attached files.

Reviewer #1: No

Reviewer #2: **Yes: **Gordon Dugle

---

## [Editor Report · Acceptance letter]

16 May 2024

PONE-D-23-41812R1 

PLOS ONE

Dear Dr. Pasquale, 

I'm pleased to inform you that your manuscript has been deemed suitable for publication in PLOS ONE. Congratulations! Your manuscript is now being handed over to our production team.

Kind regards, 

on behalf of

Dr. Claus Kadelka 

Academic Editor

PLOS ONE